# Inconsistency in the Ergogenic Effect of Caffeine in Athletes Who Regularly Consume Caffeine: Is It Due to the Disparity in the Criteria That Defines Habitual Caffeine Intake?

**DOI:** 10.3390/nu12041087

**Published:** 2020-04-15

**Authors:** Aleksandra Filip, Michal Wilk, Michal Krzysztofik, Juan Del Coso

**Affiliations:** 1Institute of Sport Sciences, The Jerzy Kukuczka Academy of Physical Education in Katowice, 40-065 Katowice; Poland; a.filip@awf.katowice.pl (A.F.); m.krzysztofik@awf.katowice.pl (M.K.); 2Centre for Sport Studies, Rey Juan Carlos University, 28942 Fuenlabrada, Spain; juan.delcoso@urjc.es

**Keywords:** classification, habituation, caffeine users, daily consumption, ergogenic aids

## Abstract

Caffeine is the most popular psychoactive substance in the world, and data suggests that it is widely used by athletes before competition to enhance physical and mental performance. The high number of athletes that regularly use caffeine suggests the need to investigate the effect of acute caffeine ingestion in athletes habituated to caffeine. However, most of the studies supporting this claim have used samples of athletes who do not consume caffeine on a regular basis, and with caffeine intake withdrawal prior to the experiments. A search through 19 databases conducted on habitual caffeine users was performed. The results of the studies regarding the ergogenic effect of caffeine in naïve vs. habitual caffeine consumers are contradictory. The diversity of results are likely associated with the use of different thresholds to categorize individuals as naïve or high caffeine consumers. There are no valid and standardized methods to accurately estimate the amount of caffeine ingested per day in athletes. We proposed a classification of athletes that habitually consume caffeine by using dietary questionnaires, and ultimately, to reduce the likelihood of discrepancies caused by the improper qualification of daily caffeine intake in studies directed at the assessment of acute caffeine intake in habitual caffeine consumers.

Nowadays, caffeine is the most popular psychoactive substance in the world and data on urine caffeine concentration suggests that it is widely used by athletes before competition to enhance physical and mental fitness [1,2]. The research supporting the ergogenic effects of acute caffeine intake in a variety of sporting disciplines and exercise scenarios is extensive, and the International Olympic Committee has recently classified caffeine as a substance that enhances performance based on this strong scientific evidence [3]. However, most of the studies supporting this claim have used samples of athletes who do not consume caffeine on a regular basis and with caffeine intake withdrawal prior to the experiments, in an attempt to enhance the margin of action of this substance on physical performance [4,5]. The high number of athletes that regularly use caffeine during training/competition suggests the need to investigate the effect of acute caffeine ingestion in athletes habituated to caffeine, because the current guidelines for caffeine use in sport may not apply to these athletes. In addition, it is vital to determine the minimal dose that enhances performance and if the prevalence of side effects is substantially modified in athletes habituated to caffeine.

After oral ingestion, caffeine is rapidly absorbed, and due to its lipophilic nature, passes through all biological membranes including the blood–brain barrier [6]. Furthermore, caffeine possesses the ability to inhibit adenosine A_1_ and A_2A_ receptors, thus reducing the fatiguing effects of this neurotransmitter during exercise [7]. Due to the blockage of adenosine receptors, caffeine indirectly affects the release of norepinephrine, dopamine, acetylcholine, serotonin, among others, which ultimately results in reducing pain, diminishing perceived exertion, and delaying fatigue [8]. However, in animal models, it has been found that chronic intake of caffeine results in more newly created adenosine receptors that partially reduce the blocking-action of caffeine on the central nervous system [9]. In fact, some reports indicate that habitual caffeine intake may change the physiological and cognitive responses to acute caffeine administration, which could negatively impact its ergogenic effect [10,11].

Several cross-sectional studies have compared the ergogenic effect of caffeine in naïve vs. habitual caffeine consumers. The results of these studies are contradictory because either naïve caffeine users benefited from 3 to 6 mg/kg of caffeine to a similar extent as habitual caffeine consumers [12,13], or naïve consumers experienced a higher ergogenic effect than habitual consumers [14]. The diversity in the results are likely associated with the use of diverse thresholds to categorize individuals as naïve or high caffeine consumers, along with differences in the exercise testing protocols. Unfortunately, evidence from well-controlled situations, in which participants underwent a standardized protocol of chronic intake of caffeine to create a potential tolerance to caffeine, is scarce and somewhat contradictory. To date, it is known that caffeine exerts benefits on aerobic and anaerobic exercise after 20 days of consecutive ingestion of 3 mg/kg/day [15]. However, the ergogenic effects of caffeine after 20 days of consecutive ingestion have been found to be lower than the effect found in the first day of ingestion when the participants are not habituated to caffeine. This suggests that there is a progressive reduction in caffeine’s ergogenicity in both aerobic and anaerobic exercise along with continuous ingestion. Interestingly, it has been found that the ergogenic effect of caffeine may be totally removed after 28 days of consecutive caffeine ingestion [16]. Despite the differences in the outcomes, two investigations [15,16] suggest that a progressive tolerance to the performance benefits of caffeine develops when this substance is ingested chronically. Interestingly, Pickering et al., [17] suggested that tolerance to caffeine’s ergogenicity could be avoided by using doses greater than the mean daily intake of caffeine. However, there are no valid and standardized methods to accurately estimate the amount of caffeine ingested per day in athletes. In athletes accustomed to daily caffeine intake, the assessment of daily intake should include dietary sources of caffeine as well as caffeine ingested as dietary supplements. Lastly, it is necessary to establish thresholds, in mg of caffeine ingested per day, to discriminate athletes habituated to caffeine from those who are naïve or low caffeine users.

Following the preferred reporting items for systematic review and meta-analyses (PRISMA) guidelines [18], we carried out a search for published studies in Medline and SportDiscus on the effects of caffeine on physical performance in subjects habituated to caffeine (search performed in April 2020). After filters were applied to remove duplicates, reviews or publications with unsuitable methodology, the search showed a total of 19 original studies that fulfilled the objective of the analysis. These investigations were selected because participants reported habitual caffeine use of >100 mg/day, the experiment contained the measurement of at least one physical performance variable, and there was a placebo/control situation. There was a restriction on the form of caffeine administration, so we discarded studies that used coffee or multi-ingredient supplements to administer caffeine in order to avoid the effect of the co-ingestion of several substances on the results of the analysis. The research procedures are presented in Figure 1.

In the final 19 studies, a total of 200 participants were catalogued as habitual caffeine users. Overall, only 17 women habituated to caffeine were included in these investigations. Interestingly, the largest number of participants habituated to caffeine under investigation was in 2019 (Figure 2), which suggests a progressive, but still insufficient interest in investigating the effects of acute caffeine ingestion in athletes habituated to caffeine.

Most of the studies on participants habituated to caffeine have assessed the ergogenic effects of acute doses of caffeine on endurance-like exercise. However, there is a lack of experiments that measure the impact of acute caffeine intake on muscular strength and other anaerobic-like variables. Only eight studies were conducted on participants with a daily caffeine intake between 100 and 299 mg/day (Table 1). Investigations within this range of daily caffeine intake are particularly interesting because this might represent an intake of ~3 mg/kg of caffeine per day for athletes with a body mass between 40 and 100 kg, whereas this is typically considered as the minimal dose that gives an ergogenic effect in naïve/low caffeine consumers [3]. 

The threshold used to classify individuals as high caffeine consumers varied from 190 mg/day [19] to more than 600 mg/day [20]. In addition, only three studies estimated daily caffeine intake in relation to body mass [21,22,23]. In those investigations that used absolute values to assess the level of habituation to caffeine, the findings could lead to incorrect conclusions because the same absolute amount of caffeine (in mg/day) would have a different impact on subjects with different body mass. Only seven studies, with a total of 95 individuals, analyzed more than seven days of consecutive caffeine intake.

All this suggests that, to date, determining the existence and the magnitude of the ergogenic effect of caffeine on habitual caffeine consumers is not feasible, particularly in women. In addition, the threshold of daily caffeine intake used to correctly categorize an individual/athlete as a habitual caffeine user has not been properly established. We conclude that the general guidelines for acute caffeine intake [3], which are based on individuals with no habituation to caffeine, are not applicable to habitual caffeine consumers because of the existence of tolerance to caffeine ergogenicity [15,16]. Before creating new caffeine guidelines for habitual users, the establishment of common and unified norms to classify individuals who ingest caffeine on a daily basis is required. The authors propose the following norms to classify individuals according to their habitual intake of caffeine (Table 2). This classification combines the most common doses of caffeine used in previous studies related to the acute effect of caffeine in elite and amateur athletes [24,25]. To correctly classify an individual, a period of at least four weeks of stable daily consumption is necessary [16,26], which should computed from both dietary sources of caffeine and from caffeine-containing dietary supplements. Therefore, the aim of this classification is to standardize the categorization of athletes that habitually consume caffeine by using dietary questionnaires, and ultimately, to reduce the likelihood of discrepancies caused by the improper qualification of daily caffeine intake in studies that aim to assess the effect of acute caffeine intake in habitual caffeine consumers. We hope that this proposal allows for a better understanding of tolerance to the performance benefits of caffeine, and allows for more precise recommendations of caffeine intake in habitual caffeine users.

The current perspective has some limitations. Studies on habitual caffeine consumption have categorized individuals as caffeine consumers by using questionnaires about dietary intake of caffeine. However, it is difficult to accurately quantify caffeine intake through dietary records due to the variations in the caffeine content in the same food products. In addition, no previous investigation has validated the data obtained through dietary questionnaires against caffeine concentration in body fluid specimens (plasma, urine or saliva). Future research that aims to validate new questionnaires for assessing habitual caffeine intake should consider assessing its validity against urine caffeine concentration (e.g., for a 24-h collection period).

## Figures and Tables

**Figure 1 nutrients-12-01087-f001:**
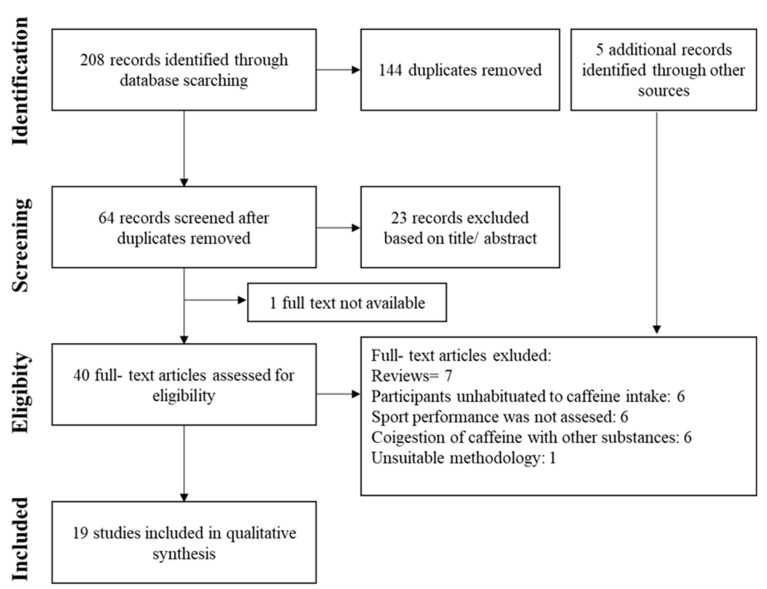
Selection of studies.

**Figure 2 nutrients-12-01087-f002:**
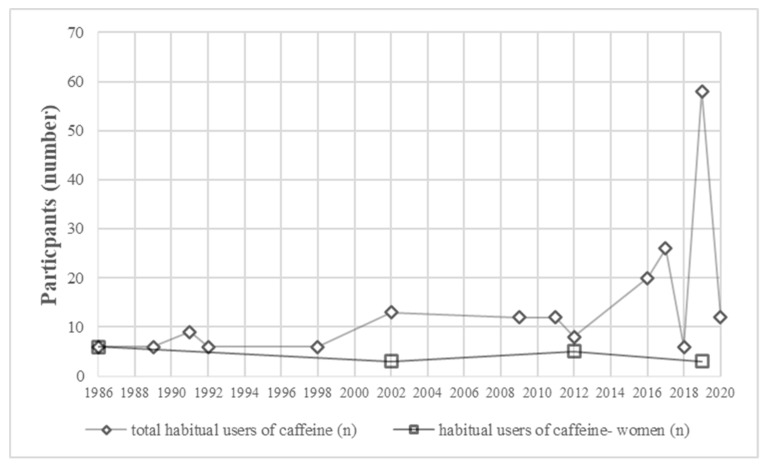
Number of participants habituated to caffeine included in research on the ergogenic effects of acute caffeine intake.

**Table 1 nutrients-12-01087-t001:** Number of participants in investigations aimed at determining the ergogenic effects of caffeine in habitual groups, depending on dose, habitual intake, type of exercise and participant’s fitness level.

		Participants
		*n* (Frequency)
Athlete level	Active	117 (58.5%)
	Trained	83 (41.5%)
	Untrained	0 (0.0%)
Type of exercise	Endurance	127 (60.2%)
	Power	33 (15.6%)
	Strength	16 (7.6%)
	Anaerobic-like	11 (5.2%)
	Speed	10 (6.6%)
	Other	14 (4.7%)
Habitual intake of caffeine	>299 mg/kg/day	123 (61.5%)
	161–299 mg/kg/day	54 (27.0.%)
	100–160 mg/kg/day	23 (11.5%)
Caffeine doses	>6 mg/kg	63 (21.2%)
	3–6 mg/kg	224 (74.9%)
	<3 mg/kg	12 (4.0%)

Studies in which participants ingested caffeine for several days to habituate them to caffeine were included in this analysis. If the experimental procedure consisted of several doses of caffeine or in several types of exercises (e.g., aerobic and anaerobic-like exercise), every experimental situation was used independently to be included in this table.

**Table 2 nutrients-12-01087-t002:** Proposed thresholds for classifying individuals in sport performance research according to their habitual caffeine consumption.

Habitual Intake of CAF	Caffeine Dose (>4wks)
Naïve consumer	<25 mg/day
Low consumer	From 25 mg/day to 0.99 mg/kg/day
Mild consumer	1.00–2.99 mg/kg/day
Moderate consumer	3.00–5.99 mg/kg/day
High consumer	6.00–8.99 mg/kg/day
Very high consumer	>9.00 mg/kg/day

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
