# Peer review of "Inconsistency in the Ergogenic Effect of Caffeine in Athletes Who Regularly Consume Caffeine: Is It Due to the Disparity in the Criteria That Defines Habitual Caffeine Intake?"

_nutrients, 2020, doi:10.3390/nu12041087_

Round 1

Reviewer 1 Report

I believe that the authors have done a good review job,
which allows us to discuss the object of study, that is,
the consumption of caffeine, habitual or not, and its effects. They present conclusions after the search, however,
I suggest the authors to include limitations of these conclusions.

Author Response

Dear Reviewer,

We are grateful for your constructive comments and suggestions that helped us to improve our manuscript. Please find below the response to your comments and the changes in the manuscript highlighted in yellow.

Reviewer 1

I believe that the authors have done a good review job, which allows us to discuss the object of study, that is,  the consumption of caffeine, habitual or not, and its effects. They present conclusions after the search, however, I suggest the authors to include limitations of these conclusions.

Reply- Thank you for appreciating our work. Yes, we agree with the Reviewer’s expert opinion and the following has been added to the text:

Page 4, line 124-131:

“The current perspective presents some limitations. Studies on habitual caffeine consumption have categorized individuals as caffeine consumers by using questionnaires of dietary intake of caffeine. However, it is difficult to accurately quantify caffeine intake through dietary records due to the variations in the caffeine content within the same food products. In addition, no previous investigation has validated the data obtained through dietary questionnaires against caffeine concentration in bodily fluid specimens (plasma, urine or saliva). Future research aimed to validate new questionnaires for assessing habitual caffeine intake should consider assessing its validity against urine caffeine concentration (e.g., for a 24-h collection period).”

On behalf of all authors, thanks for your constructive comments.

Reviewer 2 Report

Interesting topic and certainly one worthy of consideration within the sporting world. It would be great to indeed see more studies performed on habitual caffeine users and compare the results to non-habituated users, and additionally have more research studies performed within a female population, who may react differently. I do think it may be worthy to have dietary caffeine intake validated in some way through urinary measures. I was wondering if any of the studies examined had urinary measures to validate dietary measures. Could that be checked perhaps, reported on and maybe be suggested as a validation method for habitual intake. It is difficult to absolutely quantify caffeine intake through diet alone since there is such variation in caffeine content between different types of coffees, and I really do not think that has been accounted for. In addition caffeine from coffee may not be as available as caffeine from tablets, making the ergogenic effect within studies even more difficult to ascertain. Perhaps this is something which should be mentioned.

Overall the topic is really good and interesting and food for thought certainly.

Some specific comments: 

Rephrase line 40: so the result of reference 12 is clear, it is confusing to understand the exact papers result at present as, phrased.

Line 49, there is an extra space, remove it please

Line 74-76 needs to be rephrased

Line 80 - I presume CAF is short for caffeine but this is the first time it's used so needs to be in brackets and explained

Line 94 - rephrase

Author Response

Dear Reviewer,

We thank You for the constructive comments and suggestions that helped us improve the manuscript. Please find below the response to your comments and the changes in the manuscript highlighted in yellow.

Interesting topic and certainly one worthy of consideration within the sporting world. It would be great to indeed see more studies performed on habitual caffeine users and compare the results to non-habituated users, and additionally have more research studies performed within a female population, who may react differently. I do think it may be worthy to have dietary caffeine intake validated in some way through urinary measures. I was wondering if any of the studies examined had urinary measures to validate dietary measures. Could that be checked perhaps, reported on and maybe be suggested as a validation method for habitual intake. It is difficult to absolutely quantify caffeine intake through diet alone since there is such variation in caffeine content between different types of coffees, and I really do not think that has been accounted for. In addition caffeine from coffee may not be as available as caffeine from tablets, making the ergogenic effect within studies even more difficult to ascertain. Perhaps this is something which should be mentioned.

Reply - Thanks for this comment and the suggestions. Based on your comment, we have added a new paragraph that presents the limitations of previous investigations.

Page 4, line 124-131:

“The current perspective presents some limitations. Studies on habitual caffeine consumption have categorized individuals as caffeine consumers by using questionnaires of dietary intake of caffeine. However, it is difficult to accurately quantify caffeine intake through dietary records due to the variations in the caffeine content within the same food products. In addition, no previous investigation has validated the data obtained through dietary questionnaires against caffeine concentration in bodily fluid specimens (plasma, urine or saliva). Future research aimed to validate new questionnaires for assessing habitual caffeine intake should consider assessing its validity against urine caffeine concentration (e.g., for a 24-h collection period).”

Page 1, line 36-42

“Several cross-sectional studies have compared the ergogenic effect of caffeine in naive vs. habitual caffeine consumers. The results of these studies are contradictory because either naïve caffeine users benefited from 3-to-6 mg/kg of caffeine to a similar extent than habitual caffeine consumers [16,17] or naive consumers had a higher ergogenic effect than habitual consumers [18]. The diversity in the results were likely associated with the use of diverse thresholds to categorize individuals as naive or high caffeine consumers along with differences in the exercise testing protocols.”

Overall the topic is really good and interesting and food for thought certainly.

Some specific comments: 

Rephrase line 40: so the result of reference 12 is clear, it is confusing to understand the exact papers result at present as, phrased.

Reply - Done. We have changed that sentence and now reads:

Page 1,2, line:44-49

“To date, it is known that caffeine still exerts benefits on aerobic and anaerobic exercise after 20 days of consecutive ingestion of 3 mg/kg/day [12]. However, the ergogenic effect of caffeine after 20 days of consecutive ingestion were lower than the effect found in the first day of ingestion, when the participants were not habituated to caffeine, suggesting a progressive reduction in caffeine’s ergogenicity in both, aerobic and anaerobic exercise along with continuous ingestion.”

Line 49, there is an extra space, remove it please

Reply – Thanks for detecting this mistake. The change has been made

Line 74-76 needs to be rephrased

Reply - Thanks for this comment. We have rephrased that sentence:

Page 3, line: 83-87

“Only seven studies were conducted on participants with a daily caffeine intake between 100 and 299 mg/day (Table 1). Investigations within this range of daily caffeine intake are particularly interesting because this might represent an intake of ~3 mg/kg of caffeine per day for athletes with a body mass between 40 and 100 kg, while this dose is typically considered the minimal one that gives an ergogenic effect in naïve/low caffeine consumers [3].”

Line 80 - I presume CAF is short for caffeine but this is the first time it's used so needs to be in brackets and explained

Reply – We have removed the use of acronyms in the text.

Line 94 – rephrase

Reply - Thanks for this comment. We have changed to:

Page 4, line 103-105

“In addition, the threshold of daily caffeine intake to correctly qualify an induvial/athlete as habitual caffeine user has not been properly established”.

On behalf of all authors, thanks for your constructive comments.

Reviewer 3 Report

This is an interesting and well-written paper with useful implications for practise and research. As such, I only have a few minor comments that the authors may wish to consider.

Lines 56-57: What was the rationale for using >100 mg/day?

Line 74: State the actual number rather than ‘few’

Would there be any benefit to offering suggestions for definitions of low- and high-users?

I hope the authors find the above comments in the constructive manner they are intended.

Author Response

We thankful for the constructive comments and suggestions that helped us improve the manuscript. Please find below the response to your comments and the changes in the manuscript highlighted in yellow.

This is an interesting and well-written paper with useful implications for practice and research. As such, I only have a few minor comments that the authors may wish to consider.

Lines 56-57: What was the rationale for using >100 mg/day?

Reply – Thanks for this comment. We have used this threshold because the lowest daily caffeine intake used in the literature to classify and individual as habitual caffeine user was this precise value.

Line 74: State the actual number rather than ‘few’

Reply – We have included the number (“seven”).

Would there be any benefit to offering suggestions for definitions of low- and high-users?

Reply - We added a new sentence to the end of the discussion

Page 4, line: 114-118.

“Therefore, the aim of this classification is to unify the categorization of athletes habitually consuming caffeine by using dietary questionnaires, ultimately to reduce the likelihood of discrepancies among studies aimed to assess the effect of acute caffeine intake in habitual caffeine consumers due to improper qualification of daily caffeine intake.”

I hope the authors find the above comments in the constructive manner they are intended.

On behalf of all authors, thanks for your constructive comments.